# The Effects of Smart Factory Operational Strategies and System Management on the Innovative Performance of Small- and Medium-Sized Manufacturing Firms

**Rok Lee** 

Department of LINC Plus Project Organization, Gyeongsang National University, Jinju 52828, Korea;
roklee@gnu.ac.kr; Tel.: +82-10-6314-4004

**Abstract:** This study aims to determine the effects of smart factory system management and operational strategies on the innovative performance of small- and medium-sized manufacturing firms. To this end, we administered an empirical survey to 222 hands-on workers who operate smart factories in small- and medium-sized Korean firms. The collected data were analyzed using structural equation modeling and the results showed that the enterprise resource planning (ERP), quality management, ethical management, and productivity management systems had positive effects on innovative performance. The effect of operational strategies on innovative performance was not verified. Consequently, small- and medium-sized firms should focus on establishing ERP systems, which lead to system establishment, standardization of work processes, CEO support and attention, and increase user recognition levels for raising innovative performance.

**Keywords:** smart factory; strategy; innovation performance; Korea



## 1. Introduction

Workers in smart factory environments monitor the production line and, based on recorded data, perform operational management tailored to the productivity and efficiency of manufacturing the best products with the least resources. In particular, the introduction of information and communications technology (ICT) allows for quick and accurate support, and the central system of the smart factory saves time and expenses from product distribution to product release. There is a growing trend to build smart factories at the private level [1].

Such systems can be successfully built and operated by ICT when organizations integrate technology and management to create efficient system management through the adoption of smart factories. Specifically, the central government in Korea has been focusing on the adoption of 20,000 smart factories by 2022, pushing the smartification of manufacturing factories forward in a drive toward "manufacturing innovation strategies [2]".

However, although small- and medium-sized enterprises (SMEs) recognize the need to introduce smart factory systems, they are concerned about investment due to the lack of expertise regarding their introduction and cost and the technology in relation to the limited abilities of the suppliers [3].

The integration of operational strategies with the introduction of smart factories and smart factory system management in manufacturing, which will enable flexible production through the establishment and application of digital technologies in the wake of the Fourth Industrial Revolution, is recognized as inevitable. These changes provide opportunities to create new products and new business models based on data from production and sales processes [4–6].

In other words, the introduction of smart factory operational strategies and system management is inevitable, especially for SMEs aiming to become large corporations. To this end, countries such as Germany, the U.S., Japan, and China are introducing competitive

"manufacturing innovation strategies" for increasing the competitiveness of the manufacturing industry by applying digital technologies to manufacturing sites to satisfy the growing demand for smart manufacturing. This process uses digital elements from the production stage for the advancement of product characteristics and hardware types to promote embedded software and service convergence [7].

This is also an opportunity to reinvent manufacturing in Korea, as the country has come to a new understanding of the importance of SME strategic management for stable economic growth since the 2008 global financial crisis. The Fourth Industrial Revolution Committee [8] manages a high proportion of the manufacturing industry in Korea and has pointed out that there is a high risk of losing competitiveness. As such, the introduction of smart factory operation strategies and system managements was discussed as an alternative to improve innovative performance. In this study, we considered innovative performance as a dependent variable.

Further, we conjectured that enterprise resource management (ERP), quality management, ethical management, and productivity management in the manufacturing industry can improve corporate innovation performance because it is influenced by the inherent system management [9–13].

This study aims to identify the effects of smart factory system management and operational strategies, which are characteristic of Korea's SME manufacturing innovation strategies, on innovation performance by conducting an empirical survey and identifying interdisciplinary development measures and practical industry implications [14].

This study is structured as follows. Sections 2–5 cover the theoretical background, research design, results and discussion, and conclusions of the analysis, respectively.

## 2. Theoretical Background

### 2.1. Operational Strategy of a Smart Factory

The operational strategies of a smart factory can be largely summarized as facility and business automation, along with the integration of internal and external resources. The basis for the operation of a smart factory begins with automating the partial or full production of existing manufacturing facilities and connecting production facilities [15]. Wiktorsson et al. [16] stated that facility automation means unmanned facility operation through the replacement of human labor with robots or machines.

Here, facility and business automation as an operational strategy is an essential factor and affects production efficiency (as an increase) and costs (as a reduction). Additionally, it provides competitive advantages, such as rapid responses to new product development, retention of product quality and a reduction in defect rates, retention of work safety, and production reflecting customer needs.

For SMEs with insufficient resource capabilities, the integration of internal and external resources can improve competitiveness, leading to corporate innovation, which becomes a driving force for growth and development. In particular, the strategy of using government support is representative of the external operational resources of SMEs and an important factor for the successful operation of smart factories.

Therefore, this study measures and analyzes facility and business automation and the integration of internal and external resources as operational strategies for the introduction of a smart factory.

### 2.2. System Management

System management refers to a real-time management method for actively coping with environmental changes and increasing performance through the integration of systemic thinking into management activities to achieve the management goals of a successful company.

In this sense, system management is a highly efficient, autonomous management system that improves competitiveness by enhancing and then maintaining continuity to link the planning, execution, and evaluation phases systematically while also systematiz-

ing each step. It is thus an efficient management system that improves productivity by systematically standardizing and systematizing business processes [17]. There are several studies on management systems as key determinants of innovation performance [18–20].

Companies are introducing ERP systems into system management to redesign their organizational structures and innovate their information systems in response to the rapidly changing business environment. ERP is not simply a tool for establishing an information system but is accepted as part of management innovation that continuously transforms the corporate organization based on management strategies in response to changes in the business environment [21]. It also improves the reliability and efficiency of information through the unified management of business processes and data across the enterprise and helps dramatically improve management speed and productivity by identifying corporate business status in real time [22].

Under a systemic approach, the quality innovation of an organization, from product design to production and sales activities, represents the quality management system [23]. A quality management system is a systematic management basis for companies to increase innovation activities at the enterprise level based on quality management [1]. As it can satisfy the needs of customers by improving the product and quality level of a company as well as playing an important role in improving corporate competitiveness, process-oriented system management is required to satisfy customer demand [19]. An ethical management system refers to a reasonable system that is integrated and constructed so that a management program can efficiently achieve the goals and objectives of ethical management in connection with the existing management system within the organization [24]. A productivity management system refers to a management system that improves productivity by examining the productivity level and capability of the management system to increase corporate productivity [25]. It has the main purpose of designing and implementing a realizable management system through which a company can improve innovative performance.

Therefore, this study measures the management of smart factory systems for manufacturing SMEs by considering ERP, quality, ethical, and productivity management.

## 2.3. Smart Factory Operation Strategy and Innovative Performance in SMEs

There are several empirical studies on the effects of overall smart factory operation strategies on innovation performance in companies in Korea and other countries. For instance, Yam et al. [26] empirically verified the relationship between smart factory operation strategies and innovation performance in 213 innovative manufacturers in Beijing, China. Specifically, these researchers studied whether smart factory operation strategies affect the innovation performance of SMEs and showed that the effects of individual operation strategies related to each innovation performance type on innovation performance varied by company size—large, medium, or small. Although smart factory operation strategies affect all manufacturing SMEs in terms of innovation rate and performance, resource allocation showed significant results only for small companies. The frequency analysis of individual innovation performance further showed that R&D and resource allocation competences, R&D and strategic planning competences, and resource allocation and marketing competences were significant for large companies, medium companies, and SMEs, respectively.

Aghajari and Senin [27] studied Malaysian manufacturing SMEs whose operational strategies deliver innovative market outcomes, the core of which is a strategic mindset. This strategic mindset and the innovative actions of companies' management achieved operational and financially desirable results.

Lee and Jung [28] empirically determined the direct and indirect effects of technology innovation smart factory strategies and technology commercialization capabilities on management innovation performance using the mediating role of market information orientation for 183 Inno-Biz companies in Korea. They measured smart factory operation capabilities and innovation performance by dividing the former into three categories: production, marketing, and commercialization capabilities. Their results can be summarized

as follows. First, smart factory operation strategies had a positive effect on innovation performance, while regular operation strategies did not have a significant effect on innovation performance. Second, technology commercialization capabilities also had a positive overall effect on innovation performance. These results indicate that, as a result of the nature of Inno-Biz companies, innovative performance is not dependent on smart factory operation strategies but, rather, is improved only when their technological base and organizational management are efficiently integrated and operated. In developing new products and technologies, the most important aspect is the technological base. Consequently, smart factory operation strategies in manufacturing SMEs will affect innovation performance.

### 2.4. System Management and Innovative Performance of SMEs

Maas et al. [13] identified the role of ERP on organizational innovation and highlighted its importance. Karim et al. [12] stated that the scope of ERP implementation has a positive effect on business process outcomes, given ERP's substantial benefits. Xie et al. [11] surveyed China's banking companies and determined that the threat of corruption has a positive effect on new product innovation and that policy instability and competition positively improve the relationship between corruption and new product innovation. Additionally, Shafique et al. [29] suggested that ethical leadership directly influences creativity and organizational innovation, thus suggesting that business ethics are an important determinant of innovation performance.

Therefore, continuous efforts to establish quality corporate cultures through systemic approaches and process management systems are needed to improve quality levels through quality management activities [30], while process-oriented system management is necessary to provide quality at the level required by customers [19]. It is also necessary to build an ethical management system that provides excellent human resources and organizational members with ethical and moral qualities, which are basic elements of an ethical management system [20].

The scientific analysis of various determinants of corporate management and their systematic integration into management activities from the comprehensive perspective of system formation will significantly contribute to management innovation and enhance the competitiveness of SME ventures [31].

As customer demand diversifies due to the rapidly changing market environment, SME environmental management is negatively affected by the reduced product life cycle and expectations for high quality at low costs. These developments have highlighted the importance of management innovation to secure a competitive advantage in the market. If there is no innovation, the competitive advantage of a firm can be imitated and will eventually have to rely on price competition over products or services, which has become commonplace [6,32].

A company's performance over a specific period determines the success of its management. In general, performance is measured by financial performance (e.g., sales, market share, net profit growth) and non-financial performance (e.g., customer satisfaction, productivity improvement, defect reduction) [33,34].

In the study of performance, innovation performance is closely related to corporate system management, which includes the achievement of organizational goals, the capability to exploit the environment for resource acquisition, adapt to and survive the changing environment, develop human resources and satisfy the needs of organizational members, and remain productive and profitable [35]. Therefore, this study aims to identify the relationship between system management and innovation performance in SMEs.

## 3. Research Design

### 3.1. Research Model

As previously mentioned, the purpose of this study was to determine the empirical effects of smart factory operation strategies and system management on the innovation performance of SMEs based on constructs from previous studies. The independent variables

are the ERP, quality management, ethical management, and productivity management systems, which are also components of the smart factory operation strategy and system management, while the dependent variable is represented by innovation performance. These variables were analyzed using a structural equation model using the AMOS program, as per Figure 1.

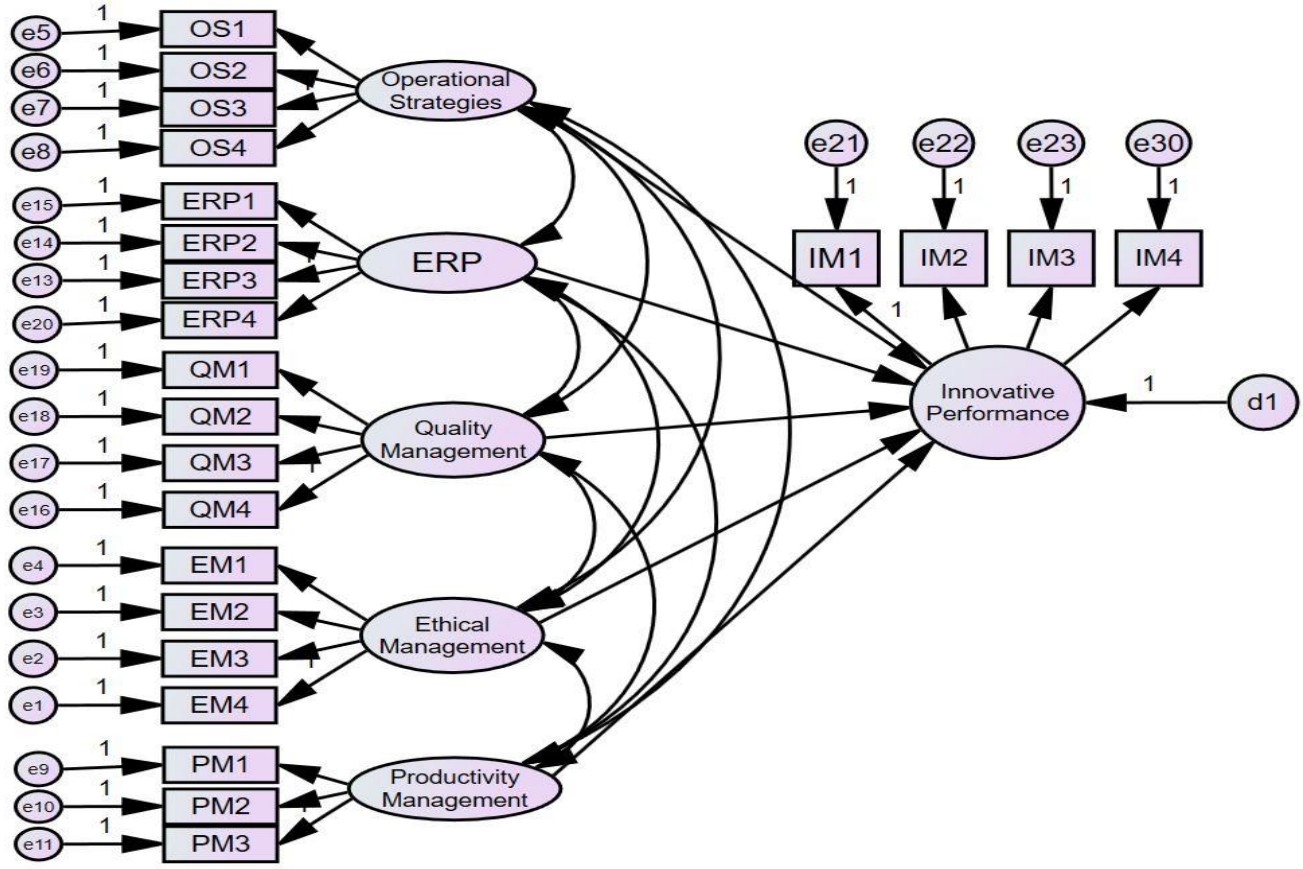

**Figure 1.** Research model.

*3.2. Questionnaire*

Table 1 shows the operational definitions of the variables and the composition of the questionnaire. The survey was conducted with hands-on workers in charge of introducing and operating SME smart factories in Korea. The survey period was from 18 May to 12 June 2020, and a structured questionnaire was used. A total of 260 questionnaires were distributed and 222 valid responses were used for the final analysis, after excluding 38 questionnaires with false or missing values. The questionnaire items were measured on a five-point Likert scale and the general characteristics were measured on a nominal scale. Please also see the Appendix A. Questionnaire.

**Table 1.** Questionnaire structure.

| Variable | | (1) Operational Definition | No. of Items | (2) Source |
|---|---|---|---|---|
| Operational strategy | | A strategy for the automation of business and production operation by using manufacturing facilities and data and the integration of resources after the introduction of a smart factory | 4 | [15] |
| System Management | ERP | A system that enables in-service workers and top managers to integrate and manage corporate resources efficiently and manage corporate resources and facilities | 4 | [36,37] |
| | Quality/System management | A system that aims to improve the product quality and management balance by improving the corporate management system through a systematic approach | 4 | [32,38,39] |
| | Ethical management | An integrated system in which the ethical management program can be linked with other existing systems in a company to achieve ethical management goals and objectives | 4 | [5,24] |
| | Productivity management | A management system to improve productivity | 3 | [25] |
| Innovative performance | | Quality and performance for price, release frequency of new products, and level of intellectual property rights | 4 | [26] |
| General information | | Industry type, number of workers, sales, positions | 4 | |
| Total | | | 27 | |

*3.3. Reliability and Validity Analyses*

Table 2 shows the reliability and validity of each variable. The smart factory operation strategy had a Cronbach's α indicating the reliability and validity of each variable. The smart factory operational strategies, system management, innovative performance in a company aims to achieve this, thus, satisfying validity requirements [40].

The ERP, quality management, ethical management, and productivity management as sub-factors of system management had Cronbach's α values of 0.7 or higher, indicating satisfactory reliability. The factor analysis of validity showed that the Kaiser–Meyer–Olkin test (KMO) value as a single factor exceeded 0.6 for all sub-factors. Innovation performance also had a Cronbach's α indicating satisfactory reliability. The reliability and the KMO value as a single factor exceeded 0.6 for each sub-factor. The factors were thus used for analysis without any modifications.

**Table 2.** Results of the validity and reliability tests.

| Factor | | Factor Loading | Eigen Value | Variance (%) | KMO [1] | Cronbach's $\alpha$ |
|---|---|---|---|---|---|---|
| Operational strategies | OS1 | 0.814 | 3.173 | 79.328 | 0.825 | 0.912 |
| | OS2 | 0.926 | | | | |
| | OS3 | 0.933 | | | | |
| | OS4 | 0.884 | | | | |
| System Management | ERP [2] | ERP1 | 0.827 | 2.524 | 63.101 | 0.792 | 0.804 |
| | | ERP2 | 0.802 | | | | |
| | | ERP3 | 0.793 | | | | |
| | | ERP4 | 0.754 | | | | |
| | Quality management | QM1 | 0.828 | 2.553 | 63.837 | 0.765 | 0.836 |
| | | QM2 | 0.799 | | | | |
| | | QM3 | 0.804 | | | | |
| | | QM4 | 0.763 | | | | |
| | Ethical management | EM1 | 0.670 | 2.713 | 67.833 | 0.790 | 0.835 |
| | | EM2 | 0.887 | | | | |
| | | EM3 | 0.870 | | | | |
| | | EM4 | 0.849 | | | | |
| | Productivity management | PM1 | 0.806 | 1.957 | 65.247 | 0.685 | 0.733 |
| | | PM2 | 0.798 | | | | |
| | | PM3 | 0.819 | | | | |
| Innovative performance | IP1 | 0.770 | 2.568 | 64.197 | 0.774 | 0.811 |
| | IP2 | 0.792 | | | | |
| | IP3 | 0.823 | | | | |
| | IP4 | 0.819 | | | | |

[1] KMO: Kaiser–Meyer–Olkin test; [2] ERP: enterprise resource management.

### 3.4. Hypotheses Development

System management can improve the reliability and efficiency of information by the unified management of business processes and data, increase management speed and productivity by understanding the corporate management status in real time, and promote quality innovation from product design to production and sales activities. Such system management is adopted to actively cope with rapid environmental changes and strengthening competitiveness. As a result, simultaneous changes in organizational structure, strategy, organizational culture, and management technique are promoted for the constitution of the entire organization [41].

An empirical analysis of the differences before and after the introduction of an ERP system using financial data from companies showed that the system had a positive effect on financial and operating performance [42]. Jung and Jung [37] conducted research on companies that had introduced and used the ERP system for more than one year, showing that the system had a positive effect on both financial and non-financial performance.

Kim and Jang [43] showed that quality management activities had a positive effect on financial performance such as operating earnings, market share, earnings rate, and sales in corporate management. Additionally, companies can achieve economic performance while fulfilling their ethical responsibilities [44] and actively practicing ethical management activities. In other words, ethical management can improve corporate competitiveness.

Further, Purwanto et al. [45] stated that the productivity management system had a significant effect on management performance, while Prakash et al. [46] conducted an empirical analysis of the relationship between the productivity management system and productivity management performance, showing that the former had a positive effect on the latter.

In accordance with these prior studies, the introduction and use of ERP systems, quality management, ethical management, and productivity management could be expected to affect innovation performance:

**Hypothesis 1.** *A smart factory operation strategy will have a significant effect on innovation performance.*

**Hypothesis 2.** *Smart factory system management will have a significant effect on innovation performance.*

**Hypothesis 2-1.** *The ERP for smart factory system management will have a significant effect on innovation performance.*

**Hypothesis 2-2.** *Quality management in smart factory system management will have a significant effect on innovation performance.*

**Hypothesis 2-3.** *Ethical management in smart factory system management will have a significant effect on innovation performance.*

**Hypothesis 2-4.** *Productivity management in smart factory system management will have a significant effect on innovation performance.*

*3.5. Data*

Table 3 shows the characteristics of companies whose employees responded to the questionnaire. By industry, these were 40 electric, electronics, semiconductor, and telecommunications companies (18.0%); 45 heavy equipment and auto parts companies (20.3%); 40 farm machinery and marine engine parts companies (18.0%); 28 metal and machinery companies (12.6%); and 69 other companies (31.1%). There were less than 50 employees in 22 companies (9.9%), 51–100 in 71 companies (32.0%), 101–200 in 115 companies (51.8%), and more than 201 in 14 companies (6.3%). Regarding sales volumes, 64 companies (28.8%), 42 companies (18.9%), 53 companies (23.9%), 28 companies (12.6%), and 35 companies (15.8%) had sales of less than KRW 5 billion, 5.1–7.0 billion, 7.1–12.0 billion, 12.1–20 billion, and above 20.1 billion, respectively. The positions of respondents showed that 139 (62.6%), 48 (21.6%), 22 (9.9%), and 13 (5.9%) were below the Deputy Section Head, Section Head and Deputy Department Head, Department Head, and Executive and other, respectively.

**Table 3.** General characteristics of the respondents.

| General Information | General Information | Frequency (N) | Percentage (%) |
|---|---|---|---|
| Industry | Electricity, electronics, semiconductor, telecommunications | 40 | 18.0 |
| | Heavy equipment, auto parts | 45 | 20.3 |
| | Farm machinery, marine engine parts | 40 | 18.0 |
| | Metal machinery | 28 | 12.6 |
| | Other | 69 | 31.1 |
| Number of workers | Below 50 | 22 | 9.9 |
| | 51–100 | 71 | 32.0 |
| | 101–200 | 115 | 51.8 |
| | Above 201 | 14 | 6.3 |
| Sales | Below KRW 5 billion | 64 | 28.8 |
| | KRW 5.1–7.0 billion | 42 | 18.9 |
| | KRW 7.1–12 billion | 53 | 23.9 |
| | KRW 12.1–20 billion | 28 | 12.6 |
| | Above KRW 20.1 billion | 35 | 15.8 |
| Position | Below Deputy Section Head | 139 | 62.6 |
| | Section Head, Deputy Department Head | 48 | 21.6 |
| | Department Head | 22 | 9.9 |
| | Executive and other | 13 | 5.9 |
| Gender | Male | 198 | 89.2 |
| | Female | 24 | 10.8 |
| Age | Thirties | 48 | 21.6 |
| | Forties | 129 | 58.1 |
| | Fifties | 32 | 14.4 |
| | Sixties | 13 | 5.9 |
| Total | | 222 | 100 |

## 4. Results

### 4.1. Confirmatory Factor Analysis

The confirmatory factor analysis was conducted using IBM SPSS 25 and AMOS 25 to identify the validity of the research model as explained by the measurement variables and to determine the overall goodness of the model fit. We obtained Chi-square = 383.220, degrees of freedom = 287, goodness of fit index RMSEA = 0.039, RMR = 0.048, AGFI = 0.903, GFI = 0.919, CFI = 0.974, which were above the minimum required values (i.e., AGFI, GFI, and CFI of 0.9 or above and RMR of 0.05 below show model fitness) [47].

The construct reliability (CR) and average variance extracted (AVE) were calculated to verify convergent validity. Convergent validity should satisfy a standardized factor load of 0.5 or more, a t-value of 1.965 or more, an AVE of 0.5 or more, and a construct reliability of 0.7 or more. Table 4 shows the results of the confirmatory factor analysis and convergent validity, in which all variables satisfy the validity criteria.

**Table 4.** Results of the confirmatory factor analysis and convergent validity.

| Variable | | | B | β | S.E. | C.R. | CR [1] | AVE [2] |
|---|---|---|---|---|---|---|---|---|
| Operational strategy | | OS1 | 0.787 | 0.743 | 0.055 | 14.363 | 0.893 | 0.678 |
| | | OS2 | 0.966 | 0.908 | 0.044 | 22.127 | | |
| | | OS3 | 1 | 0.925 | - | - | | |
| | | OS4 | 0.854 | 0.842 | 0.046 | 18.447 | | |
| System Management | ERP [3] | ERP1 | 0.898 | 0.648 | 0.106 | 8.466 | 0.823 | 0.538 |
| | | ERP2 | 1.057 | 0.788 | 0.106 | 9.927 | | |
| | | ERP3 | 0.95 | 0.709 | 0.104 | 9.166 | | |
| | | ERP4 | 1 | 0.704 | - | - | | |
| | Quality management | QM1 | 0.881 | 0.69 | 0.093 | 9.519 | 0.846 | 0.580 |
| | | QM2 | 1.164 | 0.803 | 0.106 | 10.948 | | |
| | | QM3 | 1 | 0.731 | - | - | | |
| | | QM4 | 1.117 | 0.782 | 0.104 | 10.703 | | |
| | Ethical management | EM1 | 0.759 | 0.56 | 0.089 | 8.491 | 0.844 | 0.581 |
| | | EM2 | 1.064 | 0.822 | 0.078 | 13.64 | | |
| | | EM3 | 1.082 | 0.838 | 0.077 | 13.994 | | |
| | | EM4 | 1 | 0.815 | - | - | | |
| | Productivity management | PM1 | 1.211 | 0.743 | 0.136 | 8.9 | 0.774 | 0.535 |
| | | PM2 | 0.888 | 0.598 | 0.117 | 7.571 | | |
| | | PM3 | 1 | 0.718 | - | - | | |
| Innovation performance | | IP1 | IP1 | 0.659 | 0.096 | 9.421 | 0.856 | 0.600 |
| | | IP2 | IP2 | 0.705 | 0.08 | 10.107 | | |
| | | IP3 | IP3 | 0.777 | - | - | | |
| | | IP4 | IP4 | 0.749 | 0.082 | 10.755 | | |

[1] CR: construct reliability; [2] AVE: average variance extracted; [3] ERP: enterprise resource management.

### 4.2. Discriminant Analysis

Table 5 shows the squared values of the correlation between two variables to verify the results of discriminant validity. It can be said that there is discriminant validity when the value of AVE is larger than the square of the correlation between two variables. In Table 3, the AVE value satisfied the tolerance value of 0.5 for all variables and most of the correlation squared values were lower than 0.5 when compared with Table 4, satisfying the discriminant validity criterion.

### 4.3. Path Analysis

The goodness of fit index of the research model can be summarized as follows: Chi-square = 401.040, degrees of freedom = 287, AGFI = 0.903, GFI = 0.919, CFI = 0.969, RMSEA = 0.042, and RMR = 0.048. Therefore, the model fit is acceptable. Table 6 shows the results of the structural equation model.

**Table 5.** Correlation coefficient square values.

| Variable | ERP [1] | Quality Management | Ethical Management | Productivity Management | Innovation Performance |
|---|---|---|---|---|---|
| Operational strategy | 0.242 | 0.319 | 0.456 | 0.441 | 0.253 |
| ERP | 1 | 0.268 | 0.376 | 0.144 | 0.432 |
| Quality management | | 1 | 0.408 | 0.232 | 0.319 |
| Ethical management | | | 1 | 0.433 | 0.465 |
| Productivity management | | | | 1 | 0.272 |

[1] ERP: enterprise resource management.

**Table 6.** Results of the structural equation model.

| Path | | | B | B | S.E. | C.R. | *p* |
|---|---|---|---|---|---|---|---|
| Operational strategies | → | Innovation performance | 0.007 | 0.013 | 0.036 | 0.189 | 0.85 |
| ERP | → | | 0.358 | 0.433 | 0.075 | 4.785 | *** |
| Quality management | → | Innovation performance | 0.128 | 0.18 | 0.053 | 2.425 | 0.015 * |
| Ethical management | → | | 0.303 | 0.394 | 0.064 | 4.731 | *** |
| Productivity management | → | | 0.375 | 0.248 | 0.135 | 2.78 | 0.005 ** |

*** $p < 0.001$, ** $p < 0.01$, * $p < 0.05$.

In summary, Hypothesis 1 had no significant effect, with a path coefficient of 0.013 and CR = 0.189, $p > 0.05$. Hypothesis 2-1 had a significant effect, with a path coefficient of 0.433 and CR = 4.785, $p < 0.05$. Hypothesis 2-2 had no significant effect, with a path coefficient of 0.18 and CR = 2.425, $p < 0.05$. Hypothesis 2-3 had a significant effect, with a path coefficient of 0.394 and CR = 4.731, $p < 0.05$). Hypothesis 2-4 had a significant effect, with a the path coefficient of 0.248 and CR = 2.78, $p < 0.05$.

### 4.4. Discussion

The abovementioned findings demonstrate that system management significantly affects performance improvement through technological innovation for the management of Korean manufacturing SMEs. First, the ERP for mature system management, which makes long-term survival possible, and the effort to improve quality, ethics, and productivity are important for the creation of innovation performance in addition to short-term profits. In other words, it is important to devote efforts to building and optimizing the core processes that meet mid-and long-term corporate goals in response to environmental changes in organizational operation strategies for achieving outstanding innovation performance goals. Second, since the focus is mainly on QCD (Quality, Cost, Delivery) -based production process improvement, discovery, and management even in productivity management, the innovation performance of productivity needs significant effort for organizational development, quality management, and innovation. As shown by the research results, the establishment of system management can support organization-wide decision making to achieve a strategic performance using a balanced perspective of short- and long-term external customers and internal operating processes, thus achieving technological innovation.

These results have implications for previous supporting studies [48,49], as they suggest that the utilization of applications, devices, and platform technologies in platform business is a mechanism to increase value creation through interactions between producers and consumers. Furthermore, the results confirm the findings of certain existing stud-

ies and report data that explained the causal relationship between smart factory system management and innovation performance through empirical research based on statistical analysis. In addition, it can be concluded that successful smart factory strategies for technological innovation, which are emphasized in this study, are important for innovation performance [50]. On the basis of these results, first, it is necessary to raise awareness and establish a direction for the future application and development of smart factory technologies in SMEs and to promote needs markets, commercialization models, market standards, and industry linkages for ecosystem activation. Second, the establishment of a road map and the reestablishment of a master plan for promoting smart factories in order to accelerate domestic industry–university–institute studies on technology-based platforms, device technologies and software (SW) developments, and the government's policies on commercialization development promotion. Third, an integrated model of the entire process from order to shipment will have to be applied to constantly improve smart factory solutions for SMEs in the future and cultivate manufacturing innovation for smart factory technology systems.

## 5. Conclusions

The results showed that ERP, quality management, ethical management, and productivity management as part of system management had a positive effect on the innovation performance of smart factory operation strategies in SMEs, while the operation strategy had no significant effect on innovation performance. This means that ERP, quality management, ethical management, and productivity management, as part of the smart factory operation strategy of SMEs, can promote innovation performance.

These results suggest that SMEs should promote the establishment of an ERP system to implement smart factory operation strategies and system management in order to enhance innovation performance. Productivity-oriented management, such as ethical management, that prioritizes quality, while also improving innovation performance, should be practiced.

The quality management system should enhance innovation performance by setting customer satisfaction through quality innovation as a top priority when implementing process-oriented system integration to meet customer requirements and improve customer satisfaction with quality. It is also necessary to establish an ethical management system involving such things as the enactment of an ethical code, ethical training, and ethical counseling within a company. This implies that innovation results can be achieved when organization members establish and implement an ethical management process that enables them to act in accordance with ethical standards.

Additionally, it is necessary to actively promote system management by improving productivity management through the external ethical activities of the company reflected in transparent and honest corporate activities and ethical behavior toward stakeholders.

### 5.1. Academic and Practical Implications for Corporate Management

The abovementioned research results provide important industrial implications for the management of domestic small- and medium-sized manufacturing companies. First, the establishment of mature smart factory system management to make long-term survival possible in addition to ensuring short-term profits is important for the creation of innovative performances in the small- and medium-sized manufacturing industry. In particular, the reinforcement of internal system management strategies first needs to achieve outstanding innovative performance goals. This should establish the core process through which to achieve the mid- and long-term goals. Moreover, optimized efforts between the core processes should come first to respond to environmental changes. Second, as shown in the research results, production management needs more investment in the innovation performance measurement process, because it can measure and manage the innovative performance of technology management strategies including the smart factory system development beyond the main focus of QCD-based production process improvement, customer discovery, and management.

In other words, the existing manufacturing environment of SMEs can maximize performance through their quality by analyzing and using system management for physical and environmental conditions with a system, that is, technology by application. In addition, the stable quality and suitability based on the stability of system management without errors or obstacles can improve corporate sales and competitiveness integrated with innovation performance.

*5.2. Limitations of the Research*

The results of this study have limited generalizability, as its survey sample comprised 222 hands-on workers in a smart factory operating among domestic small- and medium-sized manufacturing industries, without considering the corporate characteristics according to the operating strategy and system structure of the use variables. Therefore, further research is needed to secure representativeness through national probability samples. This would need to be expanded upon through follow-up research because it is influenced by the characteristics of each manufacturer's unique business type and capabilities in addition to smart factory operation strategies and system management factors.

**Author Contributions:** R.L. contributed to the design and implementation of the research, the analysis of the results, and the writing of the manuscript. The author have read and agreed to the published version of the manuscript.

**Funding:** This research received no external funding.

**Institutional Review Board Statement:** Not applicable.

**Informed Consent Statement:** Not applicable.

**Data Availability Statement:** Not applicable.

**Conflicts of Interest:** The authors declare no conflict of interest.

## Appendix A. Questionnaire

**Table A1.** Operational strategy and system management.

| | Evaluation Item | Strongly Agree | Agree | Neutral | Disagree | Strongly Disagree |
|---|---|---|---|---|---|---|
| Operational strategy | 1. The environmental safety of facilities and systems is ensured so that processes and finished products have no harmful effect on the environment. | | | | | |
| | 2. The automation system for each facility, process, and line of the product manufacturing process is well established. | | | | | |
| | 3. The automation system for each facility, process, and line of the product manufacturing process is well established. | | | | | |
| | 4. It stores data collected from facilities on a server, and uses cloud technologies that share as needed. | | | | | |

**Table A1.** *Cont*.

| | Evaluation Item | Strongly Agree | Agree | Neutral | Disagree | Strongly Disagree |
|---|---|---|---|---|---|---|
| ERP | 5. Your firm has built an ERP system infrastructure for continuous maintenance. | | | | | |
| | 6. Your firm is actively using the ERP system after persuading business parties and stakeholders of its importance. | | | | | |
| | 7. Your firm has improved work efficiency by using the vendor managed inventory (VMI) system. | | | | | |
| | 8. Your firm has increased productivity by using the ERP system. | | | | | |
| Quality/System management | 9. The CEO has a strong commitment to the development, implementation, and continuous improvement of the quality management system. | | | | | |
| | 10. The organization determines and secures the resources necessary for the implementation, maintenance, and continuous improvement of the quality management system. | | | | | |
| | 11. The organization determines and reviews customer requirements related to the product, as well as determining and implementing effective methods for communicating with customers. | | | | | |
| | 12. The organization monitors information related to the customer perception as to whether the quality management system meets customer requirements as a measure of performance. | | | | | |
| Ethical management | 13. Our company has a code of ethics in place to prevent corruption. | | | | | |
| | 14. The practice of punishing violations after setting standards of conduct is always followed. | | | | | |
| | 15. There are systems that allow employees to report unethical behaviors within the company. | | | | | |
| | 16. You are well informed of your company's standards of ethical behaviors to customers, suppliers, and other organizations. | | | | | |

**Table A1.** *Cont*.

| Evaluation Item | Strongly Agree | Agree | Neutral | Disagree | Strongly Disagree |
|---|---|---|---|---|---|
| Productivity management | 17. In order to improve productivity, the company is carrying out activities to eliminate waste in terms of processes and jobs at the enterprise level. | | | | | |
| | 18. The company-wide consensus is based on changes in customer demands as to the necessity of productivity improvement. | | | | | |
| | 19. Fair performance compensation for productivity improvement is provided through the consensus formed by labor-management agreement. | | | | | |

**Table A2.** Innovative performance.

| Evaluation Item | Strongly Agree | Agree | Neutral | Disagree | Strongly Disagree |
|---|---|---|---|---|---|
| Innovation performance | 1. Our company's products have recently improved in quality and performance in relation to cost compared to similar domestic competitors. | | | | | |
| | 2. The price competitiveness of our company's products are increasing in the market. | | | | | |
| | 3. Our company has been releasing new technologies and new products frequently. | | | | | |
| | 4. Our company has more intellectual property rights (patent rights, utility model rights, etc.) for new technologies or new products compared to the same industry. | | | | | |

General Information.
1. What is your gender? ① Male ② Female
2. How old are you? ( ) Years Old
3. What is your position in your company?
① Below Deputy Section Head ② Section Head, Deputy Department Head
③ Department Head ④ Executive and Other
4. What is your company's business area?
① Electricity, Electronics, Semiconductor, Telecommunications
② Heavy Equipment, Auto Parts ③ Farm Machinery, Marine Engine Parts
④ Metal Machinery ⑤ Other
5. How many employees in your company?
① Below 50 ② 51~100 ③ 101~200 ④ Above 201
6. What was the previous year's sales?
① Below KRW 5 billion ② KRW 5.1–7.0 billion
③ KRW 7.1–12 billion ④ KRW 12.1–20 billion ⑤ Above KRW 20.1 billion

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
