# Peer review of "The Effects of Smart Factory Operational Strategies and System Management on the Innovative Performance of Small- and Medium-Sized Manufacturing Firms"

_sustainability, doi:10.3390/su13063087_

Round 1
Reviewer 1 Report
The authors are advised to explain various items associated with factors namely, a1, a2...etc. It is important to know what are these items.
Reviewer 2 Report
The theme of the paper is interesting. Both the introduction and technical background are well written.
Regarding the methodology, the authors mention that they have used SEM and AMOS for the research model, but the validation is not discussed at all in the paper. The authors are just mentioning the values of the factor loading in Table 1 and Cronbach's alpha (which I guess is calculated with other program than AMOS) but none of the specific indicators for SEM, such as: RMSEA, AVE, CR, CFI, IFI, NFI, RFI - some of these indicators are discussed later on in the results section, but how about questionnaire validation?
Is "system management" a part of the Research model in Figure 1? I advise the authors to add in the paper the research model structure which results from AMOS. In this way, one can better observe the questions included in each construction.
Please define KMO.
Please replace "subject" with "respondents", they are, in fact, humans.
Please provide other descriptive statistics such as age, gender, etc to the data in Table 3.
Please add the questionnaire in the Annex of the paper.
Please revise the data in Table 4: there are 2 columns called "B" and the values in these columns are significantly different. On another note, please explain "B". Please explain also t-values.
Please use the data in Table 3 and see whether the results are the same when you use only a part of the sample made by the respondents who have certain characteristics (e.g. the same interval for the age).
Please discuss the results you have obtained through the questionnaire by providing some graphics and interpretations.
Please discuss the limitations of the study.
Author Response
03 March 2021
Prof. Dr. Marc A. Rosen
Editor-in-Chief
Sustainability
Dear Editor,
Please find enclosed my revised manuscript entitled “The Effects of Smart Factory Operational Strategies and System Management on the Innovative Performance of Small and Medium-Sized Manufacturing Firms,” which we are submitting for your consideration.
I have incorporated a number of changes based on the valuable comments by the reviewers as follows.
Reviewer 2
- The theme of the paper is interesting. Both the introduction and technical background are well written.
Response: Regarding the methodology, the authors mention that they have used SEM and AMOS for the research model, but the validation is not discussed at all in the paper. The authors are just mentioning the values of the factor loading in Table 1 and Cronbach's alpha (which I guess is calculated with other program than AMOS) but none of the specific indicators for SEM, such as: RMSEA, AVE, CR, CFI, IFI, NFI, RFI - some of these indicators are discussed later on in the results section, but how about questionnaire validation?
I thank the reviewer for pointing out this important issue. In lines 304–315, each indicator is considered to have been sufficiently explained. The confirmatory factor analysis in Chapter 4.1 explained Chi-square=383.220, degrees of freedom=287, goodness of fit index RMSEA=0.039, RMR=0.048, AGFI=0.903, GFI=0.919, and CFI=0.974.
Moreover, the path analysis in Chapter 4.3 explained Chi-square=401.040, degrees of freedom=287, AGFI=0.903, GFI=0.919, CFI=0.969, RMSEA=0.042, and RMR=0.048.
2. Is "system management" a part of the Research model in Figure 1? I advise the authors to add in the paper the research model structure which results from AMOS. In this way, one can better observe the questions included in each construction.?
Response: I thank the reviewer for this question. In lines 218–219, the previous research model was replaced with the AMOS research model. Furthermore, exogenous variables are large categories, consisting of Operational Strategies and System Management, and the sub-variables of System Management consist of ERP, Quality Management, Ethical Management, and Productivity Management.
3. Please define KMO.
Response: I thank the reviewer for providing clarity with this comment. We have revised the sentences to make them more concise. For example, the sentences. In lines 238–246, Table 1 presented in Chapter 3.1 Research Model was moved to Chapter 3.3 Reliability and Validity Analyses and modified to Table 2, where KMO was explained from the beginning.
4. Please replace "subject" with "respondents", they are, in fact, humans.
Response: Thank you for your comment. In Table 3, “subject” was replaced with “respondents.”
5. Please provide other descriptive statistics such as age, gender, etc to the data in Table 3.
Response: Thank you for your comment. In Table 3, the items of Gender and Age were added.
6. Please add the questionnaire in the Annex of the paper.
Response: Thank you for your comment. The questionnaire was added to the appendix.
7. Please revise the data in Table 4: there are 2 columns called "B" and the values in these columns are significantly different. On another note, please explain "B". Please explain also t-values.
Response: Thank you for your comment. In Table 4, B and t were misspelled. B and the t-value were modified to β and C.R., respectively.
8. Please use the data in Table 3 and see whether the results are the same when you use only a part of the sample made by the respondents who have certain characteristics (e.g. the same interval for the age).
Response: I thank the reviewer. The same analysis was conducted on a sample of participants in their 40s. The analysis results were the same.
9. Please discuss the results you have obtained through the questionnaire by providing some graphics and interpretations.
Response: I thank the reviewer for this very important feedback. In lines 337–372, section 4.4 (Discussion), the content of the discussion was modified and supplemented by comparing the results of this study with those of previous studies.
10. Please discuss the limitations of the study.
Response: Thank you. In lines 418–427, section 5.2 (Limitations of the Research), the limitations of the research were added.
I think that these revisions have helped to improve the quality of the paper. I hope that we have addressed the reviewer’s comments in a satisfactory manner.
I look forward to hearing from you at your earliest convenience.
Sincerely,
Rok Lee, Ph.D., Professor
Gyeongsang National University
501, Jinhudae-ro, Jinju, Gyeongsangnam-do, 52828, South Korea
Phone No: +82-10-6314-4004
Fax No: +82-55-772-2476
Email Address: roklee@gnu.ac.kr

Reviewer 3 Report
Author must make the following corrections in the paper:
- Author should explain better the academic contribution of the work developed. Highlighting what is innovative / original about the existing literature
- At the end of section 1 (Introduction), author must present the structure of the paper.
-Author should explain better, based on the literature, the reason for choosing the variables described in figure 1.
- In section 4.4 (Discussion), the author should compare the results obtained with the results obtained in other works described in the literature.
-Author should explain what was the criterion for choosing the companies where the questionnaires were conducted.
-Author should develop the conclusions of the work and refer in more detail to the next steps of the work
Author Response
03 March 2021
Prof. Dr. Marc A. Rosen
Editor-in-Chief
Sustainability
Dear Editor,
Please find enclosed my revised manuscript entitled “The Effects of Smart Factory Operational Strategies and System Management on the Innovative Performance of Small and Medium-Sized Manufacturing Firms,” which we are submitting for your consideration.
I have incorporated a number of changes based on the valuable comments by the reviewers as follows.
Reviewer 3
- Author should explain better the academic contribution of the work developed. Highlighting what is innovative / original about the existing literature
Response: I thank the reviewer for this very important feedback. In section 5.1. (Academic and Practical Implications for Corporate Management), the academic and practical implications of the study were modified and supplemented.
2. At the end of section 1 (Introduction), author must present the structure of the paper.
Response: I thank the reviewer. At the end of section 1 (Introduction), the structure of the paper was presented.
3. Author should explain better, based on the literature, the reason for choosing the variables described in figure 1.
Response: I thank the reviewer for providing clarity with this comment. The definition and reference of each variable according to the research purpose (model) were presented (the reason for the selection of variables is explained by the definition of variables. For specific survey items, refer to the attached appendix).
4. In section 4.4 (Discussion), the author should compare the results obtained with the results obtained in other works described in the literature.
Response: I thank the reviewer. In lines 337–372, section 4.4. (Discussion), the content of the discussion was modified and supplemented by comparing the results of this study with previous studies.
5. Author should explain what was the criterion for choosing the companies where the questionnaires were conducted.
Response: Thank you for your comment. In section 3.2. (Questionnaire), the definition and reference of each variable according to the research purpose (model) were presented (The reason for the selection of variables will be explained by the definition of variables. For specific survey items, refer to the attached appendix).
6. Author should develop the conclusions of the work and refer in more detail to the next steps of the work.
Response: Thank you. In lines 418–427, section 5.2. (Limitations of the Research), the next stage of research was suggested.
I think that these revisions have helped to improve the quality of the paper. I hope that we have addressed the reviewer’s comments in a satisfactory manner.
I look forward to hearing from you at your earliest convenience.
Sincerely,
Rok Lee, Ph.D., Professor
Gyeongsang National University
501, Jinhudae-ro, Jinju, Gyeongsangnam-do, 52828, South Korea
Phone No: +82-10-6314-4004
Fax No: +82-55-772-2476
Email Address: roklee@gnu.ac.kr

Round 2
Reviewer 2 Report
Thank you for the revised version of the paper and for addressing the reviewers' comments. Please change the possible options from: Not at All, No, Normal, Yes, So Yes in more appropriate Likert scale options. Please increase the readability of the figures within the paper.
Author Response
Dear Editor,
Please find enclosed my revised manuscript entitled “The Effects of Smart Factory Operational Strategies and System Management on the Innovative Performance of Small and Medium-Sized Manufacturing Firms,” which we are submitting for your consideration.
Reviewer 2
Thank you for the revised version of the paper and for addressing the reviewers' comments. Please change the possible options from: Not at All, No, Normal, Yes, So Yes in more appropriate Likert scale options. Please increase the readability of the figures within the paper.
Response: I thank the reviewer for this valuable feedback. In lines 544–548, Appendix. Questionnaire, the Likert scale options, change the options from : Strongly Agree, Agree, Neutral, Disagree, Strongly Disagree were added.
I think that these revisions have helped to improve the quality of the paper. I hope that I have addressed the reviewer’s comments in a satisfactory manner.
Sincerely,
Rok Lee, Ph.D., Professor
Gyeongsang National University
501, Jinhudae-ro, Jinju, Gyeongsangnam-do, 52828, South Korea
Phone No: +82-10-6314-4004
Fax No: +82-55-772-2476
Email Address: roklee@gnu.ac.kr
Reviewer 3 Report
The authors greatly improve the paper following my comments. Thus, the paper must be accepted for publication.
Author Response
Dear Editor,
I would like to thank you for providing us this opportunity to submit a revised version of our manuscript (sustainability-1127577) titled ‘The Effects of Smart Factory Operational Strategies and System Management on the Innovative Performance of Small and Medium-Sized Manufacturing Firms' to Sustainability.
I think that these revisions have helped to improve the quality of the paper. I hope that I have addressed the reviewer’s comments in a satisfactory manner.
Sincerely,
Rok Lee, Ph.D., Professor
Gyeongsang National University
501, Jinhudae-ro, Jinju, Gyeongsangnam-do, 52828, South Korea
Phone No: +82-10-6314-4004
Fax No: +82-55-772-2476
Email Address: roklee@gnu.ac.kr